# Identification of Amyloidogenic Regions in *Pseudomonas aeruginosa* Ribosomal S1 Protein

**DOI:** 10.3390/ijms22147291

**Published:** 2021-07-07

**Authors:** Sergei Y. Grishin, Ulyana F. Dzhus, Anatoly S. Glukhov, Olga M. Selivanova, Alexey K. Surin, Oxana V. Galzitskaya

**Affiliations:** 1Institute of Protein Research, Russian Academy of Sciences, 142290 Pushchino, Russia; syugrishin@gmail.com (S.Y.G.); ulya@vega.protres.ru (U.F.D.); gluktol@gmail.com (A.S.G.); seliv@vega.protres.ru (O.M.S.); alan@vega.protres.ru (A.K.S.); 2The Branch of the Institute of Bioorganic Chemistry, Russian Academy of Sciences, 142290 Pushchino, Russia; 3State Research Center for Applied Microbiology and Biotechnology, 142279 Obolensk, Russia; 4Institute of Theoretical and Experimental Biophysics, Russian Academy of Sciences, 142290 Pushchino, Russia

**Keywords:** ribosomal S1 proteins, amyloidogenic regions, toxicity, antibacterial peptides, amyloid, mass spectrometry

## Abstract

Bacterial S1 protein is a functionally important ribosomal protein. It is a part of the 30S ribosomal subunit and is also able to interact with mRNA and tmRNA. An important feature of the S1 protein family is a strong tendency towards aggregation. To study the amyloidogenic properties of S1, we isolated and purified the recombinant ribosomal S1 protein of *Pseudomonas aeruginosa*. Using the FoldAmyloid, Waltz, Pasta 2.0, and AGGRESCAN programs, amyloidogenic regions of the protein were predicted, which play a key role in its aggregation. The method of limited proteolysis in combination with high performance liquid chromatography and mass spectrometric analysis of the products, made it possible to identify regions of the S1 protein from *P. aeruginosa* that are protected from the action of proteinase K, trypsin, and chymotrypsin. Sequences of theoretically predicted and experimentally identified amyloidogenic regions were used to synthesize four peptides, three of which demonstrated the ability to form amyloid-like fibrils, as shown by electron microscopy and fluorescence spectroscopy. The identified amyloidogenic sites can further serve as a basis for the development of new antibacterial peptides against the pathogenic microorganism *P. aeruginosa*.

## 1. Introduction

The study of amyloids as ordered fibrillar protein aggregates is of great importance for elucidating their role in human pathologies, especially in neurodegenerative diseases [1,2,3]. It is known that, under certain conditions, most proteins and peptides tend not only to aggregation, but also to form amyloid-like fibrils [4,5,6]; in a particular case, the formation of amyloids of some proteins can be induced by other amyloidogenic proteins and peptides [7,8]. Currently, interest in the study of amyloids is also associated with the fact that they can be used in various nano- and bio-technological developments, including as antimicrobial agents against pathogenic microorganisms [9,10,11]. In recent reviews of scientific articles, the prospects of using antimicrobial peptides in medicine are discussed [12,13], including those acting by the mechanism of directed coaggregation with the target protein due to the interaction of amyloidogenic sites that constitute the spine of amyloid fibrils [14]. Disruption of the native structure of the most important bacterial proteins, in particular ribosomal ones, caused by directed aggregation, can be accompanied by a loss of the functional activity of the protein, which, in turn, can lead to a change in normal cellular metabolism and the death of bacteria.

The ribosomal S1 protein is the largest bacterial protein of the 30S ribosomal subunit and can perform, in addition to structural, many other functions, interacting with both RNA and other proteins [15,16,17,18]. It was shown that amber mutation and knockout of the gene encoding the bS1 protein lead to the death of bacterial cells [19,20]. The bS1 protein, which is present only in bacterial cells, contains, depending on the taxonomic affiliation of the microorganism, from one to six domains of the S1 protein (D1–D6), separated by flexible regions [21,22]. It is important that the S1 domain is a structural variant of the oligosaccharide/oligonucleotide-binding fold (OB-fold) [23,24] and can exhibit amyloidogenic properties, like another analog of the OB-fold, the cold shock domain [25]. Previously, peptides with amyloidogenic properties and antimicrobial activity against *Thermus thermophilus* were synthesized and studied based on the sequences of the S1 domains of the ribosomal S1 protein of the model organism *T. thermophilus* [26].

*P. aeruginosa* is a pathogenic bacterium that can cause nosocomial infections [27,28], and for which cases of multiple antibiotic resistance are increasingly being reported [29,30]. Recently, antimicrobial peptides have been considered as an alternative to classical antibiotics for the treatment of diseases caused by multidrug-resistant strains of *P. aeruginosa* [31,32,33]. Information about the amyloidogenic regions in the structure of the ribosomal S1 protein from *P. aeruginosa* (bPaS1) will allow the development of new antimicrobial peptides that specifically interact with this target protein and cause its aggregation, which will ultimately lead to disruption of the functioning of the ribosomal S1 protein and suppress the vital activity of this pathogenic bacteria.

The main contribution to the formation of amyloids is made by amino acid residues, which contribute to a denser packing of the protein structure [34,35]. Consequently, protein regions included in the spine of amyloid fibrils are characterized by high resistance to protease treatment, which is used to determine amyloidogenic regions in products of limited proteolysis of aggregates [36,37].

In the present work, amyloidogenic fragments were identified in the amino acid sequence of bPaS1, using the programs for searching and predicting amyloidogenic regions FoldAmyloid [38], Waltz [39], Pasta 2.0 [40] and AGGRESCAN [41], and experimentally by analyzing the products of limited proteolysis of bPaS1 aggregates using high performance liquid chromatography and mass spectrometry (LC-MS). The tendency to amyloid formation of peptides synthesized on the basis of amyloidogenic regions of bPaS1 was studied by electron microscopy (EM) and fluorescence spectroscopy (using thioflavin T (ThT)), which are widely used to detect amyloids [42,43,44].

## 2. Results

### 2.1. Isolation and Purification of bPaS1

The *E. coli* strain was obtained, the genetic construct allows us to obtain the recombinant bPaS1 with additional inserts: an N-terminal sequence with 6 His, which allows the use of affinity chromatography purification; a specific TEV protease recognition site for cleaving intact bPaS1. Nucleic acids were precipitated with streptomycin sulfate and the precipitates were removed from protein samples. The degree of purification of bPaS1 preparations was assessed by electrophoresis of samples under denaturing conditions. The resulting final preparation had a purity of at least 90%.

### 2.2. Prediction and Experimental Determination of bPaS1 Regions Prone to Aggregation

The ability of a protein to aggregate and form amyloid-like fibrils is primarily determined by the presence of amyloidogenic regions in its structure, which can be predicted using special programs developed for this purpose. Prediction of amyloidogenic sites for bPaS1 was performed using four programs: FoldAmyloid, Waltz, AGGRESCAN, and Pasta 2.0 (Figure 1B).

Each program predicts at least one region prone to amyloid formation in the bPaS1 sequence. However, the prediction results differ between different programs as they use different algorithms to find amyloidogenic regions. Subsequently, an experimental search for protein regions resistant to the action of proteases was carried out in the course of limited proteolysis and analysis of hydrolysates by LC-MS. In total, 146 significant peptides were found in the products of limited proteolysis of bPaS1 aggregates. At the same time, only 96 significant peptides were detected in the control sample without incubation for aggregation. Subsequently, significant peptides identified in the hydrolysates of control and experimental bPaS1 samples were ranked by length, the longest of them was compared with the bPaS1 sequence in order to determine the regions most protected from the action of proteases in aggregates and control preparations (Figure 1).

As shown in Figure 1B, the overall peptide coverage for protein aggregate hydrolysates and controls is similar. At the same time, additional amino acid sequences for aggregates have been identified that may play a role in the formation of associates. LC-MS data were analyzed, and peptides with a length of at least five amino acid residues were selected (similar to the selection criterion in programs predicting amyloidogenic sequences of at least five amino acid residues), which are present only in hydrolysates of aggregates and are not observed in control samples (Table 1).

As follows from Table 1, the results of bioinformatic analysis and experimental determination of amyloidogenic regions in the bPaS1 sequence do not coincide for all protease-resistant peptides found in protein aggregates. At the same time, for the identified peptides FEESLK, AIITGIVVDI, DVNGIR, LHITDMAWKR, ITDFGIFIGL, ASLHEK, KQEVESA, the accuracy of molecular weight measurement was no worse than 1.8 ppm, and the T function value was at least two times higher than the threshold value, which on the whole indicates a high reliability of the experimental determination. Thus, the bPaS1 regions that overlap with the results of predicting amyloidogenicity by at least two programs, or are identified only in the products of limited proteolysis of bPaS1 aggregates, were used as prototypes for the synthesis of peptides: AIITGIVVDI, SWIVLEAAFA, ITDFGIFIGL and LHITDMAWKR (Figure 1B). Interestingly, the local distribution of non-polar amino acid residues, especially V, I, F, C, can be used to assess the propensity of a peptide to form amyloid structures [45]. The AIITGIVVDI, SWIVLEAAFA, ITDFGIFIGL fragments are characterized by a high percentage of nonpolar amino acid residues (70%, 70%, and 60%, respectively), in contrast to the LHITDMAWKR peptide (40%).

The bPaS1 regions, which are theoretically predicted to be amyloidogenic and experimentally resistant to the action of proteases, are of interest for further study and discussion of the prospects for using antimicrobial peptides acting on the basis of directed coaggregation in the development of antimicrobial peptides.

### 2.3. Electron Microscopic Images of Aggregates

Recombinant bPaS1 was isolated, purified and analyzed using the EM method. According to EM data (Figure 2), bPaS1 under conditions of 50 mM TrisHCl, pH 8.0; 100 mM NaCl; 10 mM MgCl_2_; 5 mM β-mercaptoethanol forms disordered aggregates. That is, as in the case of the recombinant protein bS1 from *T. thermophilus* [47], bPaS1 does not form fibrils. However, it should be noted that bPaS1, in contrast to the previously studied bS1 from *T. thermophilus*, is less prone to aggregation and forms small and less dense aggregates of various sizes [47]. The images of amyloids/aggregates of peptide synthesized based on the predicted amyloidogenic regions in the bPaS1 amino acid sequence are shown in Figure 3.

According to EM data, it was shown that the AIITGIVVDI, SWIVLEAAFA, and ITDFGIFIGL peptides under conditions of 50 mM TrisHCl, pH 7.5; 150 mM NaCl, incubation for 5 h at 37 °C are able to form amyloid-like fibrils of various morphologies. Under the same conditions, the LHITDMAWKR peptide did not form fibrils, but only disordered aggregates.

### 2.4. Thioflavin T Fluorescence Assay for Aggregation of bPaS1 and Peptides

The property of thioflavin T to bind to amyloid fibrils with a simultaneous multiple increase in fluorescence at a wavelength of ~485 nm [48] was used by us to analyze the tendency towards the formation of amyloids in bPaS1 preparations and AIITGIVVDI, SWIVLEAAFA, ITDFGIFIGL, LHITDMAWKR peptides (Figure 4). In Figure 4 (Part 1), the ThT fluorescence intensity at a wavelength of ~485 nm, exceeding the control values for free ThT by a factor of ten or more, was obtained for preparations of the AIITGIVVDI, SWIVLEAAFA, ITDFGIFIGL peptides (Figure 4C–E,K, Part 1), as well as in a mixture of these peptides with bPaS1 (Figure 4G–I,L, Part 1). At the same time, for bPaS1 preparations and the LHITDMAWKR peptide (Figure 4B,F, Part 1), as well as for their mixture in solution (Figure 4J, Part 1), a multiple increase in the ThT fluorescence intensity was not observed.

Thus, the presence of the effect of a multiple increase in the ThT fluorescence intensity upon incubation with preparations of the AIITGIVVDI, SWIVLEAAFA, ITDFGIFIGL peptides and the absence of such an effect for preparations with the LHITDMAWKR peptide is consistent with the data of electron microscopy that the AIITGIVVDI, SWIVLEAAFA, ITDFGIFIGL peptides form amyloid fibrils, while only disordered aggregates of the peptide are found in the LHITDMAWKR preparations.

It should be noted that when testing the propensity for coaggregation of individual peptides with bPaS1, the greatest increase in the ThT fluorescence intensity was observed in a mixture of the ITDFGIFIGL peptide with bPaS1 after 24 h of incubation (Figure 4i, Part 2).

Thus, although bPaS1 preparations do not form amyloid-like fibrils, they affect the change in the relative intensity and wavelength of the maximum intensity of ThT fluorescence in mixtures with amyloidogenic peptides. No such effects were observed in mixtures of bPaS1 with the non-amyloidogenic LHITDMAWKR peptide.

## 3. Discussion

The tendency of bS1 proteins to form associates is determined by the structural features of the protein, as well as by the functions it performs [49,50,51]. High amyloidogenicity and, first of all, the mobility of S1 domains relative to each other creates problems in crystallization of ribosomal protein bS1 and determination of its tertiary structure [47,52]. At the moment, there is no data on the complete spatial structure of any ribosomal S1 protein. In the present work, for the isolated recombinant S1 protein from the pathogenic organism *P. aeruginosa*, regions of the amino acid sequence prone to aggregation and formation of amyloid were determined. The results obtained will be important in crystallization of bPaS1, and can also be used in the development of antimicrobial peptides acting on the basis of the mechanism of directed coaggregation with the whole protein [26].

The results of a theoretical search for amyloidogenic regions in the bPaS1 sequence using the algorithms of the FoldAmyloid, Waltz, Pasta 2.0, and AGGRESCAN programs showed that all six protein domains contain sequences prone to aggregation. Several programs predicted the same amyloidogenic regions: AIITGIVVDIDG (21–35 a.a., D1), RAESWIVLEAAFA (92–103 a.a., between D1 and D2), LHITDM (218–223 a.a., D3), ITDFGIFI (374–381 a.a., D5). At the same time, no amyloidogenic regions were identified in amino acid sequences at the N- and C-terminal fragments of the protein outside the domains. Earlier, other researchers noted that not for all proteins the prediction results of the FoldAmyloid, Waltz, Pasta 2.0, and AGGRESCAN programs coincide with the experimental data [53]. In this regard, it is of interest to discuss the data of bioinformatic analysis of amyloidogenicity in the context of the experimental determination of areas inaccessible for the action of a mixture of proteases.

The bPaS1 regions corresponding to significant unique peptides identified by limited proteolysis and LC-MS for protein aggregates only appear to be responsible for the propensity of bPaS1 to aggregate at 37 °C. Thus, the following peptides from Table 1 are common between the predicted presumably amyloidogenic and experimentally identified as resistant to proteases: AIITGIVVDI, VHAGLK, DVNGIR, LHITDMAWKR, ITDFGIFIGL. From our point of view, the AIITGIVVDI, SWIVLEAAFA, ITDFGIFIGL, LHITDMAWKR peptides were of interest for the synthesis and experimental verification of amyloidogenic properties. One should take into account both the differences in the algorithms for predicting amyloidogenic regions by the FoldAmyloid [38], Waltz [39], Pasta 2.0 [40], and AGGRESCAN [41] programs, and the possible effects of nonspecific hydrolysis of protein sequence regions by proteases, in particular, Proteinase K [54,55]. In this regard, the synthesized peptides AIITGIVVDI, SWIVLEAAFA, ITDFGIFIGL, LHITDMAWKR were also tested by other methods for detecting amyloids, namely EM and fluorescence spectroscopy with ThT.

EM studies have shown that the morphological features of the bPaS1 protein aggregates (six domains) are similar to those obtained for the *T. thermophilus* S1 protein (five domains) [47]. Despite the fact that the buffer conditions for aggregation experiments in these studies were somewhat different, it can be concluded that in both cases proteins did not form fibrils similar to amyloidogenic proteins and peptides [56,57,58], but only disordered aggregates with a tendency to form larger associates. Possibly, this morphology can be associated with the presence of several mobile domains containing amyloidogenic regions, which, when domains move relative to each other, prevents the formation of ordered fibrils [59]. In addition, as shown by us and other authors, individual protein domains exhibit different amyloidogenicity in proteins that are able to functionally interact with both nucleic acids and other proteins and peptides [52,60]. In particular, it was found that the amyloidogenicity of S1 proteins decreases with increasing protein size, while short amyloidogenic peptides synthesized based on the sequence of S1 proteins from *E. coli* and *T. thermophilus* are able to form amyloid-like fibrils [52]. In this work, using EM, it was demonstrated that three of the four selected and synthesized peptides (AIITGIVVDI, SWIVLEAAFA, and ITDFGIFIGL) were able to form amyloid-like fibrils at a temperature 37 °C and pH value 7.5, similar to the conditions for the development of the pathogenic bacteria *P. aeruginosa*.

In accordance with EM data, the amyloid-like properties of the AIITGIVVDI, SWIVLEAAFA, and ITDFGIFIGL peptides, associated with a multiple increase in the ThT fluorescence intensity, were also demonstrated using fluorescence spectroscopy. In the future, it is planned to conduct additional studies to identify the morphology of aggregates and the mechanism of formation of amyloid-like fibrils in mixtures of amyloidogenic peptides with bPaS1.

## 4. Materials and Methods 

### 4.1. Isolation and Purification of bPaS1

The DNA fragment encoding bPaS1 was amplified using KOD Hot Start DNA Polymerase (Novagen, Darmstadt, Germany), two S1P_Nde oligonucleotides (5′-CACCATATGAGCGAAAGCTTCGCAGAAC-3′) and S1P_Bam (5′-CATAGGATCCTTAGCCCTGATTCTCCATCTG-3′) and genomic DNA from *P. aeruginosa* as a DNA template. DNA fragments were digested with restriction endonucleases NdeI and BamHI (Thermo Scientific, Dreieich, Germany) and ligated with DNA of the pET19mod vector (Novagen, Madison, WI, USA), which had been previously digested with the corresponding enzymes. DNA sequencing of the pET19mod-bS1 vector was performed at ZAO Evrogen (Evrogen, Moscow, Russia). *E. coli* BL21 (DE3) cells were transformed with the pET19mod-bS1 plasmid. The cell culture was grown in LB medium in the presence of ampicillin (100 μg/mL) at 37 °C to an optical density of A590 = 0.6. Expression was induced by the addition of isopropyl-β-D-1-thiogalactopyranoside at a concentration of 0.3 mM. After induction, the cells were grown under the same conditions for 3 h. To purify the protein, the biomass of *E. coli* was suspended in 50 mM Tris-HCl buffer, pH 7.5; 10 mM MgCl_2_, 200 mM NaCl (Buffer A) containing 0.3 mM PMSF. The cells were disrupted using an ultrasonic disintegrator (Qsonica, Newtown, CT, USA). Debris was sedimented by centrifugation at 10,000× *g* for 30 min. The supernatant was loaded onto a Ni-Sepharose column (GE Healthcare, Danderyd, Sweden) equilibrated with starting buffer A. The protein was eluted with 250 mM imidazole in starting Buffer A.

LiCl and streptomycin sulfate were added to the preparation to final concentrations of 3 M and 3% (weight/weight), respectively. The mixture was incubated at +4 °C overnight. The precipitate was removed by centrifugation at 10,000× *g* for 30 min. Then the drug was transferred by dialysis into a buffer solution of 50 mM Tris-HCl, pH 7.5; 10 mM MgCl2, 50 mM NaCl and clarified by centrifugation. Then the preparation was purified by gel filtration on Superdex 75 in Buffer A. The protein preparation was stored at −20 °C. Protein concentration was determined by the Bradford method [61], as well as spectrophotometrically taking into account the extinction coefficient at 278 nm, ε_278_ = 0.75 (mL mg^−1^ cm^−1^) (determined by the method [62]). Before the experiments, aliquots of the drug (1 mg of protein) were transferred by dialysis into a buffer solution of 50 mM TrisHCl, pH 7.5; 150 mM NaCl. Protein preparations were centrifuged for 30 min (+4 °C) at 65,000× *g* in a Z 36 HK centrifuge (HERMLE Labortechnik GmbH, Wehingen, Germany). The obtained samples at a concentration of 13–16 μM (0.8–1 mg/mL) were used for experiments on electron microscopy and limited proteolysis.

### 4.2. Prediction of bPaS1 Amyloidogenic Sites

To search for amyloidogenic sites in bPaS1, we analyzed the amino acid sequence available in the open UniProt database in FASTA format under the number Q9HZ71 (RS1_PSEAE) (UniProt. Available online: https://www.uniprot.org/uniprot/Q9HZ71 (accessed on 20 January 2021)). To predict amyloidogenic sequences, we used four programs—FoldAmyloid [38], Waltz [39], Pasta 2.0 [40], and AGGRESCAN [41], designed to search for amyloidogenic regions in the protein chain.

### 4.3. Limited Proteolysis and Analysis of Hydrolysates of bPaS1 Aggregates

For experiments on the proteolysis of bPaS1 aggregates, protein preparations in a buffer solution (50 mM TrisHCl, pH 7.5; 150 mM NaCl) were divided into control and experimental samples. The former was stored at –20 °C, and the latter were incubated to obtain aggregates at 37 °C with shaking for a day at 450 rpm in a thermostatically controlled Thermomixer comfort mixer (Eppendorf, Hamburg, Germany). After that, the control and experimental samples were separately incubated with an equimolar mixture of three proteases—Proteinase K (AppliChem, Darmstadt, Germany), trypsin and chymotrypsin (Sigma-Aldrich, St. Louis, MO, USA) for 8 h at 37 °C with a shaking speed of 450 vol./min in a thermostatic mixer Thermomixer comfort (Eppendorf, Hamburg, Germany). The final concentration of bPaS1 in the preparations was 16 μM (1 mg/mL), the concentration of each protease was 0.2 μM (0.0058 mg/mL for proteinase K, 0.0048 mg/mL for trypsin; 0.005 mg/mL for chymotrypsin). After incubation, the proteolysis reaction was stopped by adding concentrated trifluoroacetic acid (1% (volume/volume)). Control and experimental samples were centrifuged for 20 min at 12,000× *g* in an Eppendorf 5418R centrifuge (Eppendorf, Hamburg, Germany). The precipitate was washed with 100 mM NH_4_HCO_3_ (pH 7.5), then all preparations were dried and dissolved in 20 μL of a solution of 0.5% (volume/volume) trifluoroacetic acid and 3% (volume/volume) acetonitrile for subsequent LC-MS analysis. To purify the samples treated with proteases, filtration was performed through ZipTips (MilliporeSigma, Burlington, MA, USA). Protein hydrolysates were concentrated on an Acclaim PepMap 100 guard column (C18, particle size 3 μm, pore size 100 Å, inner diameter 300 μm × length 5 mm) and separated on an Acclaim PepMap RSLC analytical column (particle size 2 μm, pore size 100 Å, internal diameter 75 μm × length 150 mm) (Thermo Scientific, Waltham, MA, USA) using a nano-flow liquid chromatograph EASY nLC 1000 (Thermo Scientific, Waltham, MA, USA). An Orbitrap Elite high-resolution mass spectrometer (Thermo Fisher Scientific, Dreieich, Germany) was used as a detector. Mass-to-charge (*m*/*z*) ratio ions was determined in the course of mass spectrometric analysis. The obtained *m*/*z* values, in turn, were analyzed with the PEAKS Studio 7.5 software (Bioinformatics Solution Inc., Waterloo, ON N2L 6J2, Canada) to identify peptides. A preliminary selection of identified peptides with a molecular weight > 600 Da was carried out.

### 4.4. Synthesis and Peptide Preparations for Aggregation Experiments

The sequences of the amyloidogenic regions of bPaS1, which coincide according to the predictions of at least two of the four algorithms (FoldAmyloid, Waltz, AGGRESCAN, and Pasta 2.0), were used to select four peptides (10 amino acid residues) for synthesis. Solid phase synthesis was carried out according to the Fmoc method [63]. Peptides AIITGIVVDI (1013 Da), SWIVLEAAFA (1106 Da), LHITDMAWKR (1271 Da), and ITDFGIFIGL (1095 Da) were synthesized at IQ Chemical LLC (S. Petersburg, Russia). The purified peptide was tested using an Orbitrap Elite mass spectrometer (Thermo Fisher Scientific, Dreieich, Germany). The estimated peptide molecular weight coincided with the calculated one, the purity of peptides was >95%. Immediately before the study of peptide aggregation, a stock solution was prepared with a peptide concentration of 50 mg/mL in 100% dimethyl sulfoxide (Sigma-Aldrich, St. Louis, MO, USA). To the resulting stock, solution was added a buffer solution of 50 mM Tris-HCl, pH 7.5; 150 mM NaCl to a final peptide concentration of 0.5 mg/mL. Each thus prepared peptide preparation was incubated for 5 h at 37 °C with shaking at 450 rpm in a thermostatically controlled Thermomixer comfort mixer (Eppendorf, Hamburg, Germany) and then used for electron microscopic analysis. Thus, the incubation time (5 h) and buffer solutions (with pH 7.5) for the peptides were chosen similarly to those used in [52].

### 4.5. Transmission Electron Microscopy

BPaS1 preparations (0.8 mg/mL) in buffered conditions with 50 mM TrisHCl, pH 8.0; 100 mM NaCl; 10 mM MgCl2; 5 mM β-mercaptoethanol, as well as peptides (0.5 mg/mL) in buffer conditions, 50 mM TrisHCl, pH 7.5; 150 mM NaCl was analyzed using a JEM-1200EX electron microscope (JEOL, Tokyo, Japan) according to the method used previously [46]. Before analysis, the concentration of drugs was adjusted with the appropriate buffer to 0.2 mg/mL. Samples were prepared for negative contrast analysis. A copper mesh (400 Mesh, Electron Microscopy Sciences, Hatfield, PA, USA) coated with a formvar film (0.2% (weight/volume) formvar solution in chloroform) was placed on a drop of the preparation (~10 μL). After adsorption (5 min), the meshes with the preparation were transferred to a 1% (weight/volume) aqueous solution of uranyl acetate (1 min). The analysis of the preparations was carried out at an accelerating voltage of 80 kV. The shooting was carried out on Kodak film (SO-163) for EM at a magnification of 40,000.

### 4.6. Study of bPaS1 Aggregates and Peptides Using the Fluorescent Dye Thioflavin T

Preparations of bPaS1 (1 mg/mL) and peptides (0.5 mg/mL) in buffer conditions 50 mM TrisHCl, pH 7.5; 150 mM NaCl was incubated with thioflavin T (ThT) (0.06 mg/mL) for 24 h at 37 °C with shaking for 24 h at 450 rpm in a thermostatic mixer Thermomixer comfort (Eppendorf, Hamburg, Germany). Fluorescence spectra were measured at 37 °C using an RF-6000 spectrofluorometer (Shimadzu Corporation, Kyoto, Japan) in quartz cuvettes with an optical path length of 0.3 × 0.3 cm and a volume of 100 μL of a protein and/or peptide preparation. The excitation wavelength of ThT was 450 nm, the emission spectra were recorded in the range 455–600 nm. Measurements for the same preparation were performed three times. Standard deviations were calculated for the average (based on the results of three measurements) relative fluorescence intensity of ThT at a wavelength of 485 nm, close to the maximum fluorescence intensity of amyloid-bound ThT [48,64].

## 5. Conclusions

It is known that the ribosomal S1 protein can be used as a target for the development of antibiotics against the bacterium *Mycobacterium tuberculosis*, the causative agent of tuberculosis [65,66]. In turn, the present study identified amyloidogenic sequences of the S1 protein from *P. aeruginosa* that can be used to create antimicrobial peptides capable of targeting antibiotic-resistant strains of this pathogen. Such antimicrobial peptides containing amyloidogenic regions and acting by the mechanism of directed coaggregation with the ribosomal S1 protein may be a promising variant of a new type of antibiotics directed against microorganisms that cause nosocomial and other types of infections [67,68,69].

## Figures and Tables

**Figure 1 ijms-22-07291-f001:**
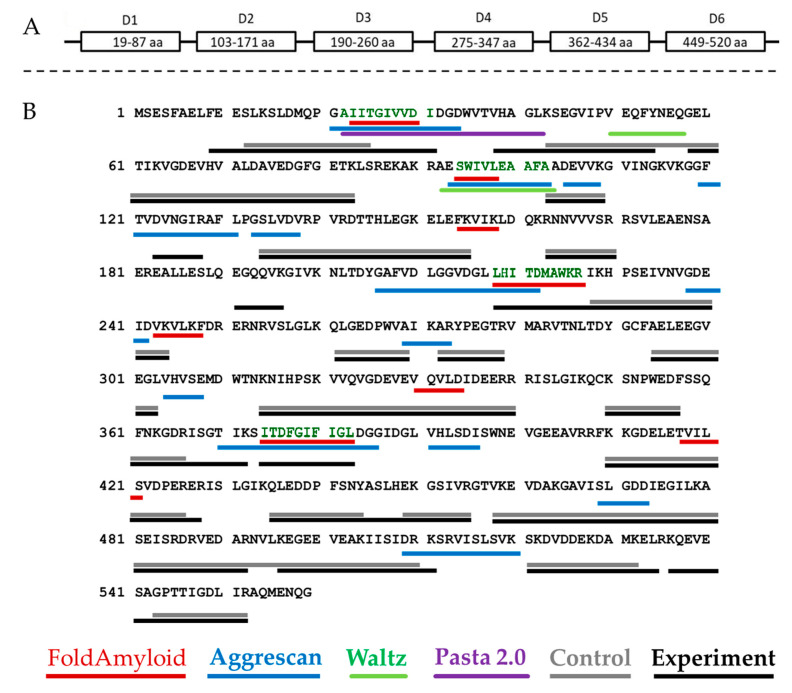
Schematic representation of the domain organization of bPaS1 (**A**) and comparison of predicting amyloidogenic regions using programs with the results of peptide coverage after LC-MS analysis of hydrolysates of control and experimental (aggregate) protein preparations (**B**). The peptides identified in the control and experimental samples, respectively, are underlined in gray and black. The bPaS1 sequence is taken from the UniProt database (UniProt. Available online: https://www.uniprot.org/uniprot/Q9HZ71 (accessed on 20 January 2021)). The regions of bPaS1, prototype peptide synthesis, are shown black–green color.

**Figure 2 ijms-22-07291-f002:**
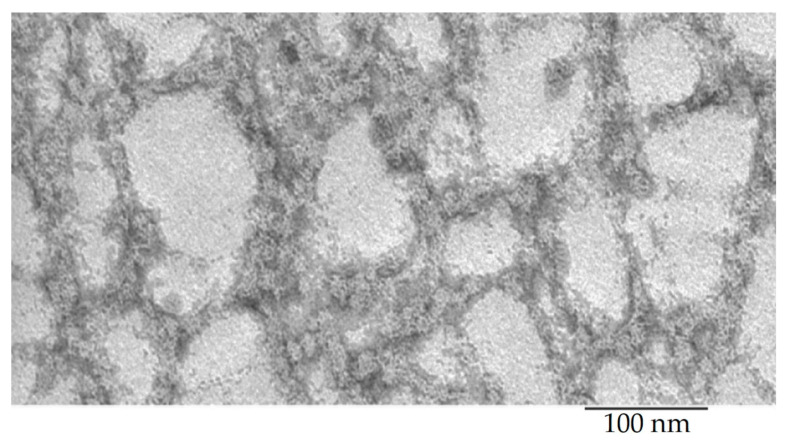
Electron microscopic image of the bPaS1 protein under conditions of 50 mM TrisHCl, pH 8.0; 100 mM NaCl; 10 mM MgCl_2_; 5 mM β-mercaptoethanol.

**Figure 3 ijms-22-07291-f003:**
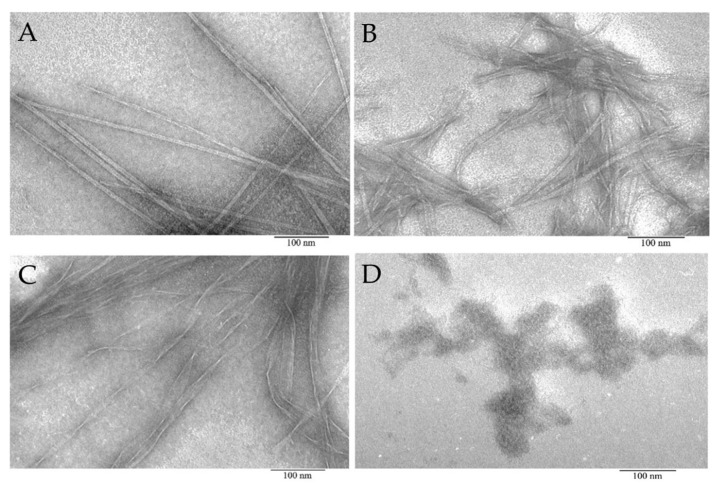
Electron microscopic images of aggregates formed from peptide preparations synthesized based on the bPaS1 sequence: AIITGIVVDI (**A**), SWIVLEAAFA (**B**), ITDFGIFIGL (**C**), and disordered aggregates of the LHITDMAWKR peptide (**D**).

**Figure 4 ijms-22-07291-f004:**
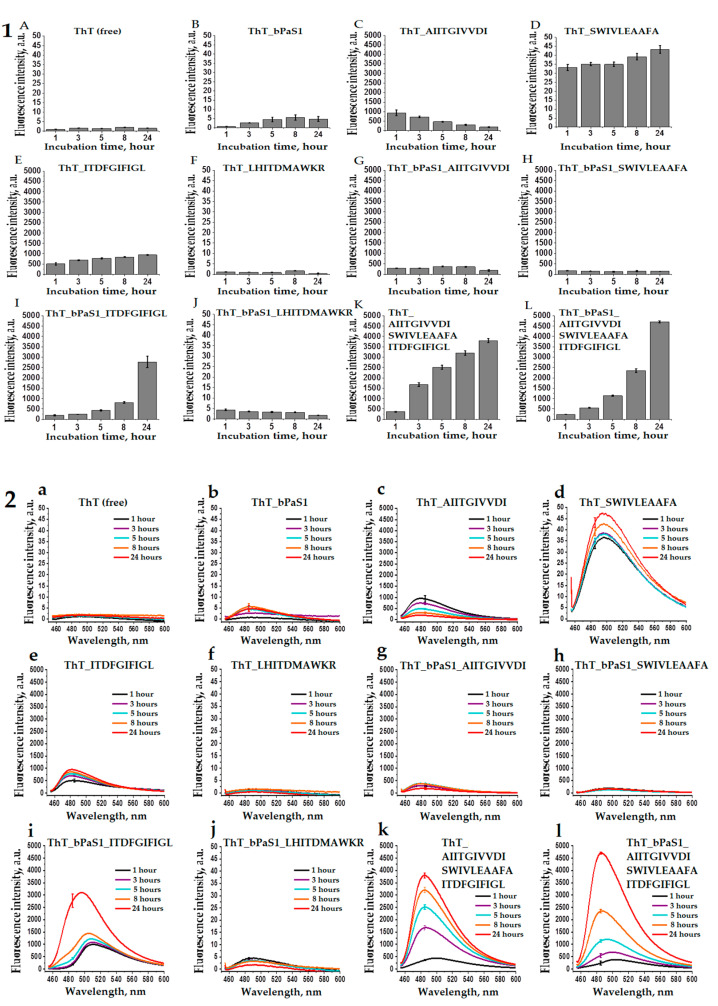
Histograms (**1**) and spectra (**2**) of fluorescence intensity of free thioflavin T (**A**,**a**) and in solution with bPaS1 (**B**,**b**), individual peptides AIITGIVVDI (**C**,**c**), SWIVLEAAFA (**D**,**d**), ITDFGIFIGL (**E**,**e**), LHITDMAWKR (**F**,**f**), a mixture of peptides (**K**,**k**), as well as in mixtures of bPaS1 with peptides (**G**,**g**), (**H**,**h**), (**I**,**i**), (**J**,**j**), (**L**,**l**). Error bars with standard deviations for the mean values of the measured fluorescence intensity after 1, 3, 5, 8, and 24 h of incubation are shown.

**Table 1 ijms-22-07291-t001:** Unique peptides identified as a result of comparing data from LC-MS analysis of hydrolysates of bPaS1 aggregates.

Peptide	Prediction of Amyloidogenicity	Percentage of Most Non-Polar a.a. [45] (V,I,F,C,L,A,M), %	Observed Mass, Da	Theoretical Mass, Da	Measurement Error, ppm *	Molecular Ion, *m*/*z*	Charge (z)	Value of the Function T **
FEESLK (9–14 a.a.)	No	0	751.376	751.3752	0.5	376.6951	+2	35.71
AIITGIVVDI (22–31 a.a.)	AGGRESCAN, Pasta 2.0, partially FoldAmyloid (23–30 a.a.)	70	1012.618	1012.6168	0.9	507.3162	+2	41.51
VHAGLK (38–43 a.a.)	Pasta 2.0	50	623.374	623.3755	–1.8	312.6945	+2	17.19
DVNGIR (123–128 a.a.)	AGGRESCAN	33	672.356	672.3555	0.7	337.1852	+2	32
E (+27.99) GQQVK *** (191–196 a.a.)	No	17	715.35	715.35	–0.3	358.6822	+2	16.8
LHITDMAWKR (218–227 a.a.)	FoldAmyloid, partially AGGRESCAN (218–223 a.a.)	40	1269.666	1269.6652	0.4	635.8401	+2	114.36
ISGTIK (367–372 a.a.)	partially AGGRESCAN (370–372 a.a.)	33	617.375	617.3748	0.7	309.6949	+2	27.5
ITDFGIFIGL (374–383 a.a.)	AGGRESCAN, partially FoldAmyloid (375–382 a.a.)	60	1094.601	1094.6012	–0.1	548.3078	+2	76.43
ASLHEK (445–450 a.a.)	No	33	683.361	683.3602	1	342.6877	+2	30.93
KQEVESA (536–542 a.a.)	No	29	789.388	789.3868	1.1	395.7011	+2	41.89

*—The accuracy of molecular weight measurement of 1 ppm (parts per million) corresponds to 0.001 Da for an ion with a molecular weight of 1000 Da. **—For the PEAKS Studio 7.5 software we used (Bioinformatics Solution Inc., Waterloo, ON N2L 6J2, Canada) the value of the function T = –10 lgP, where P is the probability that a false identification of a peptide in the current search will achieve the same or better conformity score. For peptide mapping, only peptides for which a T value > 15 were used, which corresponds to the p-criterion < 0.03 [46]. ***—Mass shift (+27.99) means amino acid post-isolation modification (formylation) at the N-termini for peptide EGQQVK.

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
