# Peer review of "Identification of Amyloidogenic Regions in Pseudomonas aeruginosa Ribosomal S1 Protein"

_ijms, 2021, doi:10.3390/ijms22147291_

Round 1

Reviewer 1 Report

High amyloidogenic potential of the ribosomal bPaS1 protein makes this molecule as a promising target for development of new microbicidal drugs. The authors of the reviewed manuscript obtained this protein in a recombinant form and using bioinformatic tools, limited proteolysis and spectrofluorimetric techniques identified several peptide motifs that, after obtaining them in a synthetic form, were able to induce aggregation of bPaS1. The formation of such insoluble aggregates and fibrils was also independently conformed by electron microscopy. The subject of the work is original, innovatory and interesting, the experiments are properly planned and executed using appropriate methods, the results are positive and well documented. However, before publication, I recommend to correct the following issues:

Major: the language of the manuscript needs substantial revision. The text contains a number of vague, ambiguous or factually improper phrases. Many sentences are too long, convoluted or difficult to understand (e.g. the lines 110-112, 137-141, 190-194, 235-240). I have found also many factual errors or ridiculous phrases (such as: “to purify bPaS1 from the impurity of nucleic acids”, “recombinant ribosomal S1 protein of Pseudomonas aeruginosa from the Escherichia coli strain”, “make up the spine”, “The results of the study of aggregates on the change in the fluorescence”, “Isolation and preparation for research of bPaS1 preparations”, “the biomass of E. coli producer bS1“, “In this regard, in this work”, “the mobility of homologous protein domains relative to each other creates problems”, “The results obtained will be important in studies of its crystallization” and other). In the present form the text is definitely not suitable for publication.

Minor comments:

  • the title of the manuscript is ambiguous. The phrases “Protein Regions from Pathogenic Bacteria” and “Pseudomonas aeruginosa Prone to Amyloid Formation” do not make sense. Please consider modification of the title into the more precise sentence, such as “Identification of amyloidogenic regions in Pseudomonas aeruginosa Ribosomal S1 Protein” or similar
  • the work contains two chapters named as Discussions (lines 83 and 204). Please correct this unusual issue
  • Figure 1: please separate the panels A and B from each other and place the letter B more appropriately (at the corner of the panel B). Please correct also the localisation of blue boxes, which overlap partially the text and the graphic elements
  • what means (+27.99) in the sequence “E(+27.99)GQQVK” presented in the Table 1? Additionally, please consider to add the column “Theoretical mass” to this Table
  • Figure 3: the legend should be corrected, the picture contain not the “images of peptide preparations” but rather the “images of aggregates formed from peptide preparations”
  • line 288, the phrase “Debris was precipitated by centrifugation” is incorrect, the debris was rather “sedimented by centrifugation”
  • line 299: the notation of the extinction coefficient as “Ԑ278” is incorrect
  • section 4.3: there is a difference between stirring and shaking. The Thermomixer apparatus from Eppendorf does not stir the samples but shakes them
  • lines 338-341: the phrases “the charge of each ion was determined, the ratio of its mass to charge (m/z). These characteristics are unique to the corresponding amino acid sequence of the peptide” are imprecise and partially incorrect
  • the abbreviation “HPLC-MS” is rarely used in literature, the more appropriate term is “LC-MS”
  • line 350: the given mass of the peptide SWIVLEAAFA is incorrectly calculated
  • section 4.4: please specify the technique used to quantitate the synthetic peptides as well as the degree of their purity. Were these peptides used in the studies without purification after synthesis?

Author Response

High amyloidogenic potential of the ribosomal bPaS1 protein makes this molecule as a promising target for development of new microbicidal drugs. The authors of the reviewed manuscript obtained this protein in a recombinant form and using bioinformatic tools, limited proteolysis and spectrofluorimetric techniques identified several peptide motifs that, after obtaining them in a synthetic form, were able to induce aggregation of bPaS1. The formation of such insoluble aggregates and fibrils was also independently conformed by electron microscopy. The subject of the work is original, innovatory and interesting, the experiments are properly planned and executed using appropriate methods, the results are positive and well documented. However, before publication, I recommend to correct the following issues:

Major: the language of the manuscript needs substantial revision. The text contains a number of vague, ambiguous or factually improper phrases. Many sentences are too long, convoluted or difficult to understand (e.g. the lines 110-112, 137-141, 190-194, 235-240). I have found also many factual errors or ridiculous phrases (such as: “to purify bPaS1 from the impurity of nucleic acids”, “recombinant ribosomal S1 protein of Pseudomonas aeruginosa from the Escherichia coli strain”, “make up the spine”, “The results of the study of aggregates on the change in the fluorescence”, “Isolation and preparation for research of bPaS1 preparations”, “the biomass of E. coli producer bS1“, “In this regard, in this work”, “the mobility of homologous protein domains relative to each other creates problems”, “The results obtained will be important in studies of its crystallization” and other). In the present form the text is definitely not suitable for publication.

Answer: We would like to thank Reviewer-1 for the constructive suggestions and comments. We have changed the text of our manuscript according to the recommendations.

Minor comments:

  • the title of the manuscript is ambiguous. The phrases “Protein Regions from Pathogenic Bacteria” and “Pseudomonas aeruginosaProne to Amyloid Formation” do not make sense. Please consider modification of the title into the more precise sentence, such as “Identification of amyloidogenic regions in Pseudomonas aeruginosa Ribosomal S1 Protein” or similar

Answer: Thanks for the valuable suggestion. The title of the manuscript has been changed to “Identification of Amyloidogenic Regions in Pseudomonas aeruginosa Ribosomal S1 Protein”.

the work contains two chapters named as Discussions (lines 83 and 204). Please correct this unusual issue

Answer: This error has been corrected.

Figure 1: please separate the panels A and B from each other and place the letter B more appropriately (at the corner of the panel B). Please correct also the localisation of blue boxes, which overlap partially the text and the graphic elements

Answer: The figure has been corrected.

what means (+27.99) in the sequence “E(+27.99)GQQVK” presented in the Table 1? Additionally, please consider to add the column “Theoretical mass” to this Table

Answer: Mass shift (+27.99) means amino acid post-isolation modification (formilation) at the N-termini for peptide EGQQVK. New column has been added to Table 1.

Figure 3: the legend should be corrected, the picture contain not the “images of peptide preparations” but rather the “images of aggregates formed from peptide preparations”

Answer: This legend has been corrected.

line 288, the phrase “Debris was precipitated by centrifugation” is incorrect, the debris was rather “sedimented by centrifugation”

Answer: We thank the Reviewer-1 for this point. We have modified the sentence.

line 299: the notation of the extinction coefficient as “Ԑ278” is incorrect

Answer: The notation has been corrected.

section 4.3: there is a difference between stirring and shaking. The Thermomixer apparatus from Eppendorf does not stir the samples but shakes them

Answer: We thank the Reviewer-1 for this remark. The mistake has been corrected.

lines 338-341: the phrases “the charge of each ion was determined, the ratio of its mass to charge (m/z). These characteristics are unique to the corresponding amino acid sequence of the peptide” are imprecise and partially incorrect

Answer: We would like to thank the Reviewer-1. We changed the paragraph.

the abbreviation “HPLC-MS” is rarely used in literature, the more appropriate term is “LC-MS”

Answer: The abbreviation has been corrected.

line 350: the given mass of the peptide SWIVLEAAFA is incorrectly calculated

Answer: We thank the Reviewer-1 for this remark. We apologize for the oversight, the mass of the peptide SWIVLEAAFA has been corrected.

section 4.4: please specify the technique used to quantitate the synthetic peptides as well as the degree of their purity. Were these peptides used in the studies without purification after synthesis?

Answer: We would like to thank the Reviewer-1. We improved the section.

Reviewer 2 Report

The study by Grishin te al. describe the potential use of  P. Aureginiosa and several of its peptides as a  potential treatment against it, mostly due to 

the formation of amyloid-like aggregates. The study employs all techniques required for the correct characterization. However, the use of the bioinformatics is substantially limited without  a chemistry assessment of their finding. I would 

sincere appreciate the authors can include a chemistry-based analysis of the polarity of the amino acid in the region of the peptide fragments and it’s comparison within the other computational tools, highlighting the most approach that render a high degree of polarity in the sequence. 

Ref. Johansson J, Nerelius C, Willander H, Presto J. Conformational preferences of non-polar amino acid residues: an additional factor in amyloid formation. Biochem Biophys Res Commun. 2010 Nov 19;402(3):515-8. doi: 10.1016/j.bbrc.2010.10.062. Epub 2010 Oct 28. PMID: 20971069.

Typos: Pag. 3 programsas 

Table 1 add molecular polar definition

Author Response

The study by Grishin te al. describe the potential use of  P. Aureginiosa and several of its peptides as a  potential treatment against it, mostly due to the formation of amyloid-like aggregates. The study employs all techniques required for the correct characterization. However, the use of the bioinformatics is substantially limited without  a chemistry assessment of their finding. I would sincere appreciate the authors can include a chemistry-based analysis of the polarity of the amino acid in the region of the peptide fragments and it’s comparison within the other computational tools, highlighting the most approach that render a high degree of polarity in the sequence. 

Ref. Johansson J, Nerelius C, Willander H, Presto J. Conformational preferences of non-polar amino acid residues: an additional factor in amyloid formation. Biochem Biophys Res Commun. 2010 Nov 19;402(3):515-8. doi: 10.1016/j.bbrc.2010.10.062. Epub 2010 Oct 28. PMID: 20971069.

 Answer: We thank the Reviewer-2 for this suggestion. Information about the non-polarity of the amino acid in the region of the peptide fragments has been added in the text and Table 1.

 Typos: Pag. 3 programsas 

Answer: This error has been corrected.

Table 1 add molecular polar definition

Answer: We would like to thank Reviewer-2 for valuable suggestions. We have changed the text of Table 1 according to the recommendations.